# SPT: Learning to Selectively Insert Prompts for Better Prompt Tuning

**Wei Zhu**[1][*] **Ming Tan**[2]
[1] East China Normal University
[2] Southern University of Science and Technology

## Abstract

Prompt tuning prepends a soft prompt to the input embeddings or hidden states and only optimizes the prompt to adapt pretrained models (PTMs) to downstream tasks. The previous work manually selects prompt layers which are far from optimal and failed to exploit the potential of prompt tuning. In this work, we propose a novel framework, Selective Prompt Tuning (SPT), that learns to select the proper prompt layers by inserting a prompt controlled by a learnable probabilistic gate at each intermediate layer. We further propose a novel bi-level optimization framework, SPT-DARTS, that can better optimize the learnable gates and improve the final prompt tuning performances of the learned prompt layer settings. We conduct extensive experiments with ten benchmark datasets under the full-data and few-shot scenarios. The results demonstrate that our SPT framework can perform better than the previous state-of-the-art PETuning baselines with comparable or fewer tunable parameters.

## 1 Introduction

Increasingly large pre-trained models (PTMs) (Han et al., 2021; Devlin et al., 2019; Peters et al., 2018; Liu et al., 2019b; Radford and Narasimhan, 2018; Raffel et al., 2019; Zhu, 2021b; Guo et al., 2021; Zuo et al., 2022; Sun et al., 2020) have achieved the state-of-the-art (SOTA) performances on most NLP tasks. Full-model fine-tuning is one of the most widely used method for utilizing PTMs. However, fine-tuning (Devlin et al., 2019; Zhu et al., 2023b, 2021a,b; Zhu, 2021a; Gao et al., 2023; Zhang et al., 2023a) needs to tune all parameters of PTMs for each task, resulting in large GPU memory and storage costs, especially for supersized PTMs (Brown et al., 2020; Wang et al., 2021a). Parameter-efficient tuning (PETuning) is a new fine-tuning paradigm that can reduce the adaptation costs of PTMs by only tuning a very small

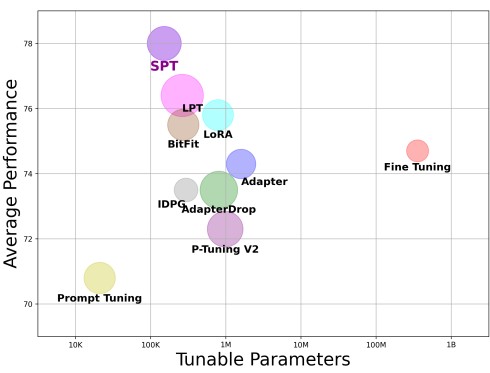

Figure 1: Overall comparison between our SPT method and baselines under the few-shot scenario with 100 training samples for each task. All methods are evaluated on ten text classification tasks using RoBERTa-large. The radius of every circle indicates training speed (tokens per millisecond).

number of internal or additional parameters (Ding et al., 2022; Zhang et al., 2023b; Zhu et al., 2023a).

Prompt tuning (Lester et al., 2021) is a simple and popular PETuning method that prepends a sequence of soft prompt tokens to the input sequence and only tunes the prompts to adapt the PTM backbones to downstream tasks. Despite its advantages in parameter efficiency and convenience in PTM deployment, prompt tuning suffers from lower performance and convergence rate than other PETuning methods like Adapters (Houlsby et al., 2019; Pfeiffer et al., 2021; Mahabadi et al., 2021; Zhang et al., 2023b), and BitFit (Ben-Zaken et al., 2021). Recently, there has been a branch of literature investigating the advanced techniques for improving the performances of prompt tuning. P-tuning v2 (Liu et al., 2021) improves the performance of prompt tuning by inserting soft prompts into every hidden layer of PTMs. However, it is difficult to optimize and needs more training steps to attain competitive performance. BBT (Sun et al., 2022) optimizes the inserted prompts with derivative-free optimization.

---

[*]Corresponding author: michaelwzhu91@gmail.com

IDPG (Wu et al., 2022) employs a prompt generator with parameterized hyper-complex multiplication (Le et al., 2021) to generate instance-aware soft prompts. LPT (Liu et al., 2022a) inserts an instance-aware late prompt generated by the neural prompt into an intermediate layer of the PTM instead of the input layer or all layers. Liu et al. (2022a) can achieve competitive performance under both full-data and few-shot scenarios. Note that the above methods adopt heuristic strategies to determine where to insert prompts on the PTMs.

In this paper, we first conduct a pilot experiment to show that simple modifications to the prompt inserting strategies in Liu et al. (2022a,b) can result in better performances than these baselines with comparable tunable parameters. The pilot experiments demonstrate that there is a dire need for the optimal strategy of setting prompt layers into PTMs. Predictably, such an optimal strategy may vary across tasks or PTM backbones and is difficult to construct manually. Therefore, we propose the **S**elective **P**rompt **T**uning (SPT) framework (Figure 2), which automatically searches for the optimal strategy of inserting prompts into the PTMs.

Our SPT framework considers a simple search space of whether to insert the generated instance-aware prompts into an intermediate layer of the PTM. As depicted in Figure 2, we initialize a prompt hyper-network where each intermediate layer of PTMs inserts a prompt controlled by a learnable probabilistic gate $\alpha_m$. We follow the bi-level optimization strategy of Liu et al. (2019a) to optimize the learnable probabilistic gates. After optimization, we keep the top prompt layers that receive the highest probabilities to meet the tunable parameter budgets. To better optimize the learnable gates and obtain better prompt layer settings, we propose SPT-DARTS, which consists of two novel techniques to improve the optimization process of Liu et al. (2019a). Our SPT framework can automatically determine the suitable prompt-inserting strategy that achieves a high-quality trade-off between parameter efficiency and model performance.

Extensive experiments are conducted on six benchmark datasets from the GLUE benchmark and four widely studied text classification benchmarks. The results show that SPT performs comparable to or outperforms the previous SOTA PETuning methods. Especially in the few-shot scenario with 100 training samples, SPT outperforms the

PETuning baselines by a clear margin. Figure 1 depicts the overall comparison between our SPT method and baselines.

To summarize, our contributions are:

- We propose the SPT framework, which automatically learns to insert instance-aware prompts at the proper intermediate layers of PTMs.

- We propose SPT-DARTS, which contains two novel techniques to improve the optimization process of the prompt hyper-network.

- We verify our SPT framework in the full-data and few-shot scenarios across ten benchmark text classification tasks and three different PTM backbones.

## 2 Related work

### 2.1 Prompt-based tuning

A major research line of PETuning is the prompt-based tuning that inserts some additional soft prompts into the embeddings or hidden states on specific layers of PTMs. Prompt tuning (Lester et al., 2021) and P-tuning (Liu et al., 2021) insert a soft prompt on the word embedding layer only and can achieve competitive results when applied to large-scale PTMs. Prefix tuning (Li and Liang, 2021) and P-tuning v2 (Liu et al., 2021) insert prompts to every hidden layer of PTM. (Zuo et al., 2022) propose to utilize prompt tuning in the continual learning of rumor detection. BBT (Sun et al., 2022) optimizes the inserted prompt with derivative-free optimization. Recently, there have been a few works investigating instance-aware prompting. IDPG (Wu et al., 2022) uses a prompt generator to generate a soft prompt for every instance. Context tuning (Tang et al., 2022) uses the pretrained BERT model (Devlin et al., 2019) as the prompt generator and focuses on NLG tasks. IPL (Jin et al., 2022) first calculates relevance scores between prompt tokens and inputs, then uses the scores to re-weight the original prompt tokens. However, IPL needs to tune all parameters of PTM. LPT inserts an instance-aware late prompt into the middle intermediate layer of the PTM instead of the embedding layer or all the Transformer layers and achieves competitive performances despite its simple design.

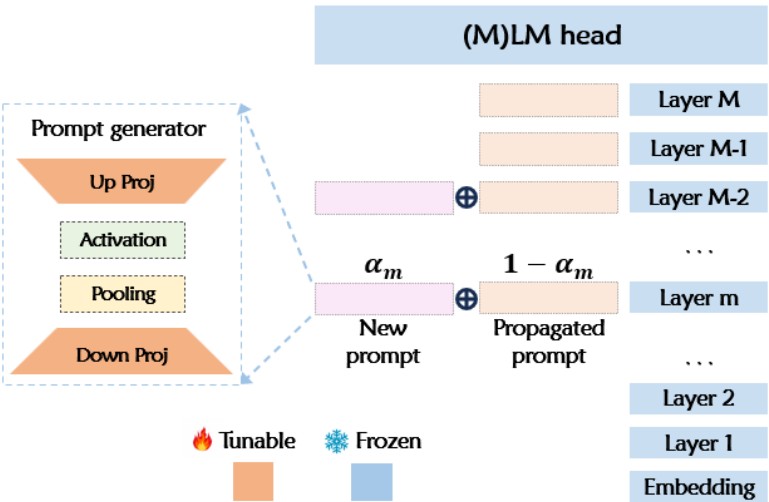

Figure 2: The overview of our SPT framework. **Left**: a prompt generator with a bottleneck architecture. **Right**: The forward pass of SPT. At each transformer layer of PLM, one has to decide whether to use the prompt propagated from the previous layer or inject a newly generated prompt.

Our work contributes to the literature by proposing SPT, which selectively inserts prompts on certain intermediate layers of PTMs and effectively boosts prompt tuning performance while maintaining high parameter efficiency.

## 2.2 Other PETuning method

One important research line of PETuning is the adapter-based tuning (Ding et al., 2022) that inserts certain adapter modules between or around the self-attention or feed-forward modules of the Transformer layer and only tunes these adapters in downstream training for model adaptation. Adapter (Houlsby et al., 2019) inserts adapter modules with bottleneck architecture between every consecutive Transformer (Vaswani et al., 2017) sublayers. AdapterDrop (Rücklé et al., 2020) improves efficiency by removing adapters from lower layers. Compacter (Mahabadi et al., 2021) used low-rank optimization and parameterized hypercomplex multiplication (Le et al., 2021) to compress adapters. Zhang et al. (2023b) propose to optimize the adapter architectures in order to obtain better fine-tuning performances. Adapter-based tuning methods have comparable results with model tuning when training data is sufficient but work less well in the few-shot scenario (Wang et al., 2021b). There are also some other popular PETuning methods, such as BitFit (Ben-Zaken et al., 2021) which only tunes the bias terms, LoRA (Hu et al., 2021) which optimizes low-rank decomposition matrices of the weights within self-attention layers.

Recently, there are work conducting automatic configurations of PETuning modules, such as Hu et al. (2022); Zhang et al. (2023b). Compared to , we focus on the prompt layer selection, which is not included in the search space of Hu et al. (2022). Thus our work can be seen as a meaningful complement to the existing literature.

## 3 Problem Formulation

For PTM full fine-tuning, the input samples are usually reformulated as [CLS] $\langle S_1 \rangle$ [SEP] if the inputs are single sentences, and as [CLS] $\langle S_1 \rangle$ [SEP] $\langle S_2 \rangle$ [SEP] if the inputs are sentence pairs. After the PTM backbone encodes the inputs, the final hidden states of the [CLS] token will be used to predict classification labels with a linear classification head.

In the settings of prompt tuning, the downstream tasks are reformulated as masked language model tasks to close the gap between pre-training and fine-tuning. Specifically, we insert randomly initialized soft prompt $p$ on the word embeddings, and also modify the original inputs using different manually designed templates with a [MASK] token for task adaptations. For example, in single-sentence tasks, the input will be transformed into a template like

$$\text{concat}(p, \text{E}([\text{CLS}] \langle S_1 \rangle \text{ It was [MASK]. [SEP]}))$$

where $\text{E}(x)$ means to map the tokens in the input sequence $x$ into embedding vectors. Then, we map the original labels $\mathcal{Y}$ to some words (label words) in the vocabulary $\mathcal{V}$ of $\mathcal{M}$. Then the final hidden

states of [MASK] token will be fed into the pre-trained masked language modeling (MLM) head to predict label words. During downstream task tuning, the PTM backbone and the MLM head will be frozen, and only the soft prompt $p$ will be tuned. This way, the downstream tasks are formulated as a masked language modeling task to close the gap between pre-training and downstream task tuning.

In the setting of our proposed SPT framework (depicted in Figure 2), we investigate the problem of whether to insert newly generated instance-aware prompts at the word embeddings or certain intermediate layers of PTM. For convenience, we will refer to the word embedding layer as the 0-th layer of the PTM. We refer to the layers at which new prompts are inserted as the prompt layers (**PLs**). At a certain prompt layer $i$, we will use a prompt generator $\mathbf{PG}_i$ to generate a prompt $\mathbf{p}_i$ from a given input's hidden states at layer $i$.

# 4 SPT: Selective Prompt Tuning

In this section, we will elaborate on our Selective Prompt Tuning (SPT) framework, which is depicted in Figure 2. We first discuss our motivations. Then we will elaborate on our method for determine the prompt layers.

## 4.1 Motivation

We have conducted a pilot experiment on the RTE (Dagan et al., 2005) and Subj (Pang and Lee, 2004) datasets in which we manually design a series of strategies to set the prompt layers of RoBERTa-large (Liu et al., 2019b). The details are put in Appendix A due to limited length. The experimental results demonstrate that: (a) simple manually designed strategies of inserting prompts could perform comparably with a recent strong baseline prompt tuning method Liu et al. (2022a), with comparable numbers of tunable parameters. (b) setting too many prompt layers will instead hurt the tuning performances. The above observations raise a vital research question which we will address:

**R.G.1**: *how do we find the optimal strategy of prompt injection, given the task at hand?*

## 4.2 prompt generators

A prompt generator is a simple feed-forward layer with a bottleneck architecture (Wu et al., 2022). It first down-projects the hidden states $\mathbf{h}$ of PTM from dimension $d$ to dimension $m$

($m \ll d$) via a linear layer $\mathrm{MLP}_{down}$. Then it obtains the prompt $\mathbf{p}$ with length $l$ through average pooling operation $\mathrm{Pooling}()$. The pooled prompt will go through an activation function $g$ and be up-projected to dimension $d$ via another linear layer $\mathrm{MLP}_{up}$.

$$\mathbf{p} = \mathrm{MLP}_{up}(g(\mathrm{Pooling}(\mathrm{MLP}_{down}(\mathbf{h})))). \quad (1)$$

Following Mahabadi et al. (2021); Wu et al. (2022), we employ the parameterized hyper-complex multiplication (PHM) layer (Le et al., 2021) with parameter $n$ to reduce the parameters of $\mathrm{MLP}_{down}$ and $\mathrm{MLP}_{up}$. PHM substitutes the weight matrix of a linear layer to a sum of Kronecker products, thus having a parameter complexity of $\mathcal{O}(md/n)$, reducing the parameters of the projection layers by at most $\frac{1}{n}$.

## 4.3 Prompt hyper-network

We aim to search for the optimal setting of prompt layers under the limited tunable parameter budgets. Assume the parameter budget allows $K$ prompt layers. Since not all prompt layers contribute equally to task performance, only a fraction of layers should be selected as prompt layers to avoid redundancy of the tunable parameters.

Thus, we initialize a prompt hyper-network where the embedding layer and all the intermediate layers have a prompt generation layer controlled by a *learnable probabilistic gate*. Introducing a zero-initialized learnable parameter $\alpha_i \in \mathbf{R}$, the learnable gate at layer $i$ is given by

$$a_i = \mathrm{Sigmoid}(\alpha_i), \quad (2)$$

where $\mathrm{Sigmoid}()$ is the Sigmoid activation function. $a_i \in (0, 1)$ can be seen as the probability of activating the prompt generator at layer $i$. At each layer of the hyper-network, the prompt $\mathbf{p}_i$ consists of the prompt $\mathbf{p}_i^{(prev)}$ propagated from the previous layer, and the prompt $\mathbf{p}_i^{(new)}$ generated from the prompt generator $\mathbf{PG}_i$ at layer $i$. Formally, the prompt $\mathbf{p}_i$ at layer $i$ is given by

$$\mathbf{p}_i = (1 - \tau * a_i) * \mathbf{p}_i^{(prev)} + \tau * a_i * \mathbf{p}_i^{(new)}, \quad (3)$$

where $\tau \in \{0.5, 1.0\}$ is a hyper-parameter determining whether to discard the previous layer's prompt $\mathbf{p}_i^{(prev)}$ when a new prompt is generated at layer $i$. Note that $\tau = 1.0$ is similar to Liu et al. (2021) where instance-independent new prompts are inserted at each intermediate layer.

Through optimization, the probabilistic gate $a_i$'s value will move toward 0 or 1, acting as importance scores for the prompt layers. The top $K$ layers that receive the highest probabilistic gate values will be set as prompt layers to meet the parameter budget, and the model with such a group of prompt layers will be referred to as the learned SPT model.

Our hyper-network, which is the backbone model with soft prompts at each layer and the prompts are controlled by the learnable gates $\alpha_i$. The parameters $\alpha_i$ are learnt jointly with the model parameters. So they are not hyper-parameters and do not require hyper-parameter tuning to determine their values.

### 4.4 Optimization of prompt hyper-network

Following DARTS (Liu et al., 2019a), we consider all the parameters $\alpha_i$ of the learnable probabilistic gates as architectural parameters, denoted as $\alpha$, and optimize them via bi-level optimization. Denote the hyper-networks' prompt generator parameters as $\omega$. The bi-level optimization optimize $\alpha$ conditioned on the optimized parameters of prompt generators $\omega^*$. At each epoch, the train set is split into two splits $\mathcal{D}_\alpha$ and $\mathcal{D}_\omega$. The inner and outer levels of optimization are conducted on these two separate splits, which is analogous to validating architectures trained on $\mathcal{D}_\omega$ using a different split $\mathcal{D}_\alpha$ to avoid over-fitting. Thus the optimization objective is:

$$
\begin{aligned}
&\min_\alpha \mathcal{L}(\mathcal{D}_\alpha, \omega^*, \alpha), \\
&s.t. \ \ \omega^* = \arg\min_\omega \mathcal{L}(\mathcal{D}_\omega, \omega, \alpha),
\end{aligned} \quad (4)
$$

where $\mathcal{L}()$ is the objective function on a given downstream task. The above bi-level optimization problem is approximated with an alternating optimization strategy. The gradients of the prompt generators are calculated with batches of samples from $\mathcal{D}_\omega$, and the gradients of $\alpha$ are calculated on $\mathcal{D}_\alpha$.

Although DARTS is widely applied, it is known to produce unstable gradients and sub-optimal performances (Dong and Yang, 2020). We propose two novel techniques to improve the optimization of architectural parameters $\alpha$. We will refer to our modifications to DARTS as SPT-DARTS.

**Re-parameterization of probabilistic gates** Note that probabilistic gates $a_i$ is calculated by equation 3. Their optimization does not explicitly consider the trade-offs across different layers,

thus not optimizing to fully reveal the difference in their contributions to the prompt hyper-network. We now introduce a re-parameterization step to $a_i$ before the calculation of equation 3:

$$
\hat{a}_i = a_i * C = a_i * \frac{\sum_i \mathrm{GD}(a_i)}{\sum_i a_i}, \quad (5)
$$

where $\mathrm{GD}()$ detaches the parameter from the computational graph, and the parameter will never have gradients. The above equation does not change the value of $a_i$ since $C$ has a value of 1. And equation 3 becomes

$$
\mathbf{p}_i = (1 - \tau * \hat{a}_i) * \mathbf{p}_i^{(prev)} + \tau * \hat{a}_i * \mathbf{p}_i^{(new)}. \quad (6)
$$

Now the gradient of $\alpha_i$ is given by:

$$
\begin{aligned}
\frac{\partial \mathcal{L}}{\partial \alpha_i} &= \sum_k \frac{\partial \mathcal{L}}{\partial \hat{a}_k} \frac{\partial \hat{a}_k}{\partial \alpha_i} \\
&= C * \frac{\partial \mathcal{L}}{\partial \hat{a}_i} \frac{\partial a_i}{\partial \alpha_i} - \sum_k a_k \frac{\partial \mathcal{L}}{\partial \hat{a}_k} \frac{\sum_i \mathrm{GD}(a_i)}{(\sum_i a_i)^2} \frac{\partial a_i}{\partial \alpha_i} \\
&= \frac{\partial a_i}{\partial \alpha_i} * \left( \frac{\partial \mathcal{L}}{\partial \hat{a}_i} - \sum_k \frac{a_k}{\sum_j a_j} \frac{\partial \mathcal{L}}{\partial \hat{a}_k} \right). \quad (7)
\end{aligned}
$$

We can see that our re-parameterization technique introduces an extra term $\sum_k \frac{a_k}{\sum_j a_j} \frac{\partial \mathcal{L}}{\partial \hat{a}_k}$ in the gradient. This way, we explicitly introduce the interactions among the gating parameters from different layers during gradient computations.

**Architectural consistency learning** Note that the final optimized model we want is sparse, with most layers' prompt generators being pruned. To close the gap between the hyper-network and the final model, we assign a Bernoulli distributed random mask $m_i \in \{0, 1\}$ with mean value $s \in (0, 1)$ to each learnable probabilistic gate $a_i$. Thus, equation 6 becomes

$$
\mathbf{p}_i = (1 - m_i * \tau * \hat{a}_i) * \mathbf{p}_i^{(prev)} + m_i * \tau * \hat{a}_i * \mathbf{p}_i^{(new)}. \quad (8)
$$

Now we ask the same input $x$ to go through the forward pass twice, once with the architectural masks applied (Equation 8) and once with the architectural masks turned off (Equation 6), resulting in different hidden representations $h_x^{(1)}$ and $h_x^{(2)}$ for the input sample. We now introduce a consistency regularization objective in addition to the task's objective function:

$$
\mathcal{L}_c = \mathbf{MSE}(h_x^{(1)}, h_x^{(2)}), \quad (9)
$$

where **MSE** is the mean squared error loss function. Note that this regularization term will be added to both the inner and outer objectives in Equation 4.[1]

Our consistency regularization objective is inspired by the recent works in consistency learning (Liang et al., 2021). Here, we apply the idea of consistency learning to enhance the optimization process of the learnable probabilistic gates. Intuitively, this regularization term encourages the hyper-network to output consistent hidden states when different sets of prompt generators are pruned. It ensures that each prompt generator is well-trained and bridges the gap between the hyper-network and the final discretized SPT model. As a result, the optimization of $a_i$ can better reflect the contributions of each prompt generator, and thus the final learned model will obtain better performance.

## 5 Experiments

### 5.1 Evaluation datasets

We evaluate our method on five single-sentence and five sentence-pair classification tasks, including six tasks from GLUE benchmark (Wang et al., 2018) and four other popular tasks, including MPQA (Wiebe et al., 2005), MR (Pang and Lee, 2005), Subj (Pang and Lee, 2004), and TREC (Voorhees and Tice, 2000) tasks. All details about the dataset statistics and evaluation metrics can be found in Appendix B.

### 5.2 Experiment Settings

All experiments are conducted on NVIDIA GTX A40 GPUs. We use Pytorch (Paszke et al., 2019) and HuggingFace's Transformers (Wolf et al., 2020) libraries to implement our SPT method. We evaluate our method in both full-data and few-shot scenarios on three PTM backbones, RoBERTa-large (Liu et al., 2019b), DeBERTa-large (He et al., 2020), and GPT2-large (Radford et al., 2019). Unless otherwise specified, the number of prompt layers $K$ is set to 4, the prompt length $l$ is 10, and we set $\tau = 0.5$ and the coefficient $\lambda_c$ of the consistency regularization term in Eq 9 to 1.0. That is, our method will keep the previous layer's prompt when inserting new prompts. Moreover, we report the average performances and standard deviations on the test set of the learned SPT model across 5

random seeds under the full-data scenario and 10 random seeds under the few-shot scenario. More implementation details are provided in Appendix C.

### 5.3 Baselines

We compare our SPT framework with the current SOTA baseline methods.

**Fine-tune** The traditional fine-tuning method that trains all parameters in the PTM backbone.

**Adapter-based tuning** we compare with (1) Adapter (Houlsby et al., 2019); (2) AdapterDrop (Rücklé et al., 2020).

**Prompt-based tuning** For prompt-based tuning methods, we compare with (1) Prompt Tuning (Lester et al., 2021), (2) P-tuning v2 (Liu et al., 2022b), (3) IDPG (Wu et al., 2022), and (4) LPT (Liu et al., 2022a). The prompt length for all these methods are set to 10.

**Other PETuning methods** We also compare: (1) BitFit (Ben-Zaken et al., 2021); (2) LoRA (Hu et al., 2021); (3) $S^3$ by Hu et al. (2022).

We implement Adapter, AdapterDrop, BitFit, and LoRA using OpenDelta[2] library. Other baselines are implemented using their open-sourced codes. For a fair comparison, we do not use supplementary training like Wu et al. (2022) to enhance performance.

### 5.4 Results in the few-shot scenario

We first evaluate our SPT framework in the few-shot scenario. Following Wu et al. (2022); Liu et al. (2022a), we consider three settings where the training set size is 100, 200, and 500, respectively. Under a given random seed, we randomly sample the training samples from the original training set. For every baseline and our SPT method, we will run the experiments over 10 different random seeds and report the mean and deviation on each task. The development and test sets are the same as the full-data scenarios.

The results for the few-shot scenario of 100 samples are presented in Table 1. The results for the few-shot scenario of 200 and 500 samples are in Table 12 of Appendix E. Our SPT method outperforms all the baseline methods in the few-shot settings. Especially when the training set has only 100 samples, the SPT method outperforms model tuning by 3.3 points on average and Adapter by 3.7 points. Our method also outperforms all the

---

[1]We set the coefficient $\lambda_c$ of this term to 1.0. Without further hyper-parameter tuning, this regularization can already help improve the model performances.

[2]https://github.com/thunlp/OpenDelta

| Method | Tunable Params | SST-2 (acc) | MPQA (acc) | MR (acc) | Subj (acc) | TREC (acc) | MNLI (acc) | MRPC (acc and f1) | QNLI (acc) | QQP (acc and f1) | RTE (acc) | Avg |
|---|---|---|---|---|---|---|---|---|---|---|---|---|
| Model tuning | 355M | 87.6 (1.2) | 80.5 (2.0) | 82.5 (2.5) | 88.6 (1.5) | 89.3 (1.9) | 51.5 (3.3) | 77.3 (1.0) | 71.9 (6.6) | 69.9 (2.1) | 48.6 (3.0) | 74.7 |
| Adapter | 1.6M | 88.3 (1.3) | 80.8 (3.0) | 82.9 (1.5) | 88.7 (0.8) | 88.7 (1.7) | 47.9 (1.2) | 76.8 (1.4) | 68.5 (2.7) | 67.3 (1.8) | 53.1 (2.4) | 74.3 |
| AdapterDrop | 811K | 86.8 (1.2) | 80.3 (2.3) | 83.3 (1.1) | 88.3 (1.2) | 88.9 (2.2) | 45.2 (0.8) | 76.4 (0.9) | 67.4 (3.9) | 65.7 (1.7) | 51.4 (1.8) | 73.5 |
| BitFit | 273K | 89.1 (0.9) | 82.0 (2.1) | 83.1 (1.0) | 87.3 (1.0) | 89.7 (1.8) | 51.0 (1.9) | 78.4 (1.7) | 69.3 (6.5) | 69.7 (0.9) | 55.8 (1.2) | 75.5 |
| LoRA | 788K | 88.5 (1.3) | 82.3 (1.3) | 83.5 (0.9) | 88.6 (1.4) | 89.9 (0.8) | 51.3 (2.7) | 77.8 (1.7) | 69.9 (5.7) | 70.3 (1.3) | 56.3 (2.0) | 75.8 |
| $S^3$ | 293k | 89.2 (1.2) | 82.5 (2.3) | 83.4 (0.8) | 89.1 (1.3) | 89.8 (1.5) | 51.8 (1.7) | 78.3 (1.3) | 70.2 (4.6) | 70.6 (1.1) | 56.9 (1.5) | 76.2 |
| Prompt Tuning | 21K | 87.1 (2.2) | 75.5 (3.8) | 82.1 (1.2) | 82.6 (2.8) | 81.3 (3.7) | 46.3 (1.8) | 74.2 (1.3) | 62.8 (2.3) | 59.7 (2.1) | 56.6 (2.3) | 70.8 |
| P-tuning v2 | 985K | 87.8 (0.6) | 78.6 (1.6) | 81.6 (2.1) | 87.7 (1.4) | 84.1 (3.1) | 41.3 (1.8) | 75.2 (1.6) | 66.2 (3.3) | 66.7 (3.0) | 53.8 (2.1) | 72.3 |
| IDPG | 296K | 88.6 (1.7) | 77.5 (5.8) | 82.7 (1.8) | 86.6 (1.5) | 85.6 (2.7) | 48.8 (1.3) | 76.1 (1.6) | 68.6 (3.1) | 64.5 (1.6) | 55.7 (2.8) | 73.5 |
| LPT | 263K | 89.7 (0.8) | 82.8 (1.4) | 83.3 (1.5) | 89.7 (1.7) | 89.3 (1.8) | 52.5 (2.0) | 78.1 (2.0) | 71.6 (1.7) | 70.8 (1.9) | 57.1 (3.5) | 76.5 |
| **Our SPT method** | | | | | | | | | | | | |
| SPT (ours) | 152K | **90.8** (1.0) | **84.5** (1.6) | **84.3** (0.5) | **90.8** (0.9) | **90.5** (1.8) | **54.9** (1.7) | **79.2** (1.5) | **73.2** (2.2) | **72.3** (1.3) | **58.9** (2.3) | **78.0** |
| SPT ($\tau = 1.0$) | 152k | 90.1 (1.1) | 83.3 (1.5) | 83.6 (1.1) | 89.6 (1.2) | 89.4 (2.2) | 53.1 (2.1) | 77.9 (1.9) | 72.1 (2.5) | 71.4 (1.1) | 57.5 (2.1) | 76.8 |

Table 1: Results in the few-shot scenario of 100 training samples. We report mean and standard deviation of performance across 10 random seeds. Bold and Underline indicate the best and the second best results. All the results are obtained using RoBERTa-large.

prompt-based baseline methods with comparable or less tunable parameters. The results demonstrate that our method can achieve good generalization performance when the training data is very scarce.

From Table 1 and 12, we can see that although outperforming the prompt tuning-based baseline methods, SPT with $\tau = 1$ generally performs less well than SPT with $\tau = 0.5$. This result is intuitive. The prompt propagated from the previous layers carries different semantic or syntactic information (Clark et al., 2019), which can help the current and future layers to better encode the input sample.

### 5.5 Results in the full-data scenario

The overall comparison of our SPT framework and the baselines in the full-data scenario is reported in Table 11 in Appendix D. From the experimental results, we can see that our SPT method can outperform the PETuning baselines with comparable or fewer tunable parameters. We can also observe from Table 11 that: (a) Generally, the prompt-based methods are weaker than the adapter-based methods under the full-data settings, especially on sentence-pair tasks, which is consistent with the results from Sun et al. (2022). However, our method overcomes this shortcoming by properly setting the prompt generators at certain intermediate layers. (b) Our method SPT with the learned prompt layer setting is comparable with or outperforms the strong baselines, like AdapterDrop, BitFit, and LoRA, with even fewer tunable parameters.

### 5.6 Analysis and ablation studies

**Visualization and discussions of the learned SPT models**  We visualize the learned SPT models on the ten tasks with RoBERTa-large backbone via heat map, as depicted in Figure 3. In Figure 3, the

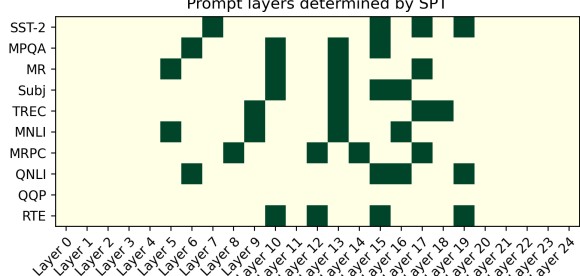

Figure 3: The heat map representing the chosen prompt layers by our SPT method on each task, using RoBERTa-large as backbone. For each cell, dark green represents a prompt layer, while bright yellow means not.

rows represent different tasks, and the columns correspond to the layer indices. For each cell, dark green represents a prompt layer, while bright yellow means not. We can observe the following patterns: (a) all the tasks decide to insert prompts after the embedding layer (layer 0) and the first four transformer layers, which is a similar observation to (Liu et al., 2022a). (b) Layers 10 to 19 of RoBERTa-large are frequently chosen as the prompt layers. Similarly, Liu et al. (2022a) observe that the middle intermediate layers are the most performing prompt layers. (c) SPT discards the last four layers. We hypothesize that if we set prompt layers among these layers, the newly generated prompt will not be propagated long enough to formulate useful task-related information.

**The effects of the number of prompt layers**  In the main experiments (Tables 1 and 11), we mainly set the number of prompt layers $K$ to 4. To investigate how $K$ affects SPT's performance, we now run the SPT method with $K \in \{1, 2, 8, 16\}$. We adjust $n$ so that different settings of $k$ have comparable tunable parameters. The results of the learned

| Settings | Tunable Params | SST-2 (acc) | MNLI (acc) | RTE (acc) | Subj (acc) |
|---|---|---|---|---|---|
| SPT ($K = 1$) | 149K | 94.9 (0.2) | 88.6 (0.2) | 76.6 (1.1) | 94.1 (0.3) |
| SPT ($K = 2$) | 150K | 94.9 (0.1) | 88.8 (0.4) | 78.1 (1.5) | 94.6 (0.2) |
| SPT ($K = 4$) | 152K | 95.2 (0.1) | **89.0** (0.3) | **79.2** (1.1) | **95.3** (0.1) |
| SPT ($K = 8$) | 157K | **95.3** (0.2) | 88.9 (0.2) | 78.9 (1.2) | 95.1 (0.2) |
| SPT ($K = 16$) | 166K | 95.0 (0.1) | 89.0 (0.2) | 78.5 (1.0) | 95.2 (0.1) |

Table 2: Results of SPT on four tasks with different numbers of prompt layers $K$. The pretrained model is RoBERTa-large. Bold indicates the best result among PETuning methods.

| Source | Target | | | |
|---|---|---|---|---|
| | SST-2 | MNLI | RTE | Subj |
| SST-2 | 95.2 (0.1) | 88.9 (0.2) | 79.1 (1.4) | 95.0 (0.3) |
| MNLI | 95.1 (0.1) | 89.0 (0.3) | 78.9 (0.8) | 94.9 (0.2) |
| RTE | 95.0 (0.2) | 88.7 (0.3) | 79.2 (1.1) | 94.7 (0.3) |
| Subj | 94.9 (0.2) | 88.8 (0.1) | 78.6 (0.6) | 95.3 (0.1) |

Table 3: Evaluation of the prompt layer settings transferred from source datasets to target datasets under the full-data scenario. The target datasets are in the column names, and the source datasets are in the row names.

| Method | Tunable Params | SST-2 (acc) | MNLI (acc) | RTE (acc) | Subj (acc) |
|---|---|---|---|---|---|
| **DeBERTa-large backbone** | | | | | |
| Model tuning | 406M | 95.7 (0.1) | 89.9 (0.2) | 83.5 (0.7) | 96.4 (0.2) |
| Prompt tuning | 26K | 94.0 (0.3) | 86.1 (0.4) | 62.8 (1.5) | 92.9 (0.4) |
| LPT | 263K | 94.9 (0.2) | 89.1 (0.3) | 78.1 (1.3) | 94.7 (0.3) |
| SPT | 152K | **95.4** (0.2) | **89.5** (0.3) | **81.8** (1.1) | **95.7** (0.4) |
| **GPT2-large backbone** | | | | | |
| Model tuning | 774M | 95.6 (0.2) | 89.1 (0.3) | 77.8 (1.2) | 96.2 (0.3) |
| Prompt tuning | 31K | 93.8 (0.2) | 86.2 (0.3) | 59.7 (2.0) | 93.0 (0.4) |
| LPT | 329K | 94.7 (0.3) | 89.1 (0.4) | 75.2 (1.6) | 94.1 (0.3) |
| SPT | 186K | **95.3** (0.2) | **89.3** (0.3) | **77.1** (1.5) | **95.2** (0.4) |

Table 4: Results on 4 GLUE tasks using DeBERTa-large and GPT2-large models as the backbone under the full-data scenario. Bold indicates the best result among PETuning methods.

SPT models can be found in Table 2. The results show that: (a) our main setting $K = 4$ performs comparable to or better than other settings of larger $K$, showing that we can not achieve performance bumps simply by adding more prompt layers. (b) Note that by learning the placement of a single prompt layer, our SPT ($K = 1$) model performs comparable to or better than the strong baseline Liu et al. (2022a). The results demonstrate that our method indeed has the ability to discover the proper prompt layers.

**The effects of prompt length** In the main experiments (Table 1), the prompt length $l$ for our SPT method and LPT, IDPG is set to 10 following LPT (Liu et al., 2022a). The same prompt length on the three methods ensures that the comparisons of the three methods are fair. Now, we change $l$ to 5 or 20, and the performances on 4 tasks are reported in Table 14 of Appendix G. From the results, we can see that: (a) our method can consistently outperform the baseline method under different prompt lengths. (b) Our method is less affected by the prompt length hyper-parameter, which is important to real-world application since increasing the prompt length increases computations quadratically.

**Transferability of the learned PG settings** We now evaluate the transferability of the learned SPT models. In table 3, we select four datasets, SST-2, MNLI, RTE, and Subj, and treat them as the source or target dataset. We search the prompt layer setting on the source dataset and train with the learned prompt layer on the target task. We can see from Table 3 that the transferred prompt layer settings can perform close to the directly learned settings and already achieve better performances than most of the baseline models. The transferability guarantees the re-usability of our SPT framework.

**Working with other pre-trained encoders** To demosntrate that our method's superiority does not rely on a specific pre-trained backbone, we run our SPT method and baselines on the DeBERTa-large

(He et al., 2020) and GPT2-large (Radford et al., 2019) backbones. The results are reported in Table 4. The results show that our method works well on the two backbones and successfully outperform the baselines.

**Training efficiency of the SPT framework** Compared with LPT (Liu et al., 2022a), our optimization framework consumes 4~5 times training time and 1.6 times GPU memory due to bi-level optimization and multiple forward passes. However, considering that the training is done off-line, it is affordable compared to manually designing different prompt layer settings and running numerous evaluations.

**Inference efficiency** We run inference on the RTE test set, with three different tasks: prompt tuning, LPT, and the learned SPT model, with batch size 32 and maximum length 128. The memory footprint and speed are recorded in Table 13 (Appendix F). We can see that during inference, all three methods consume almost equal GPU memory sizes, and SPT is 3.1% slower than LPT. The results show that our SPT method achieves superior performances while still being efficient.

## 5.7 Results on large language models

To demonstrate that our method can generalize to larger language models, we now conduct the

| Method | Tunable Params | SST-2 (acc) | MNLI (acc) | RTE (acc) | Subj (acc) |
|---|---|---|---|---|---|
| RoBERTa-large + LPT | 263K | 89.7 (0.8) | 52.5 (2.0) | 57.1 (3.5) | 89.7 (1.7) |
| RoBERTa-large + SPT | 152K | 90.8 (1.0) | 54.9 (1.7) | 58.9 (2.3) | 90.8 (0.9) |
| LlaMA-13b + LPT | 1310K | 90.6 (0.6) | 52.9 (2.3) | 57.9 (3.0) | 90.8 (1.5) |
| LlaMA-13b + SPT | 672K | 91.3 (0.8) | 55.7 (1.9) | 59.6 (2.1) | 91.2 (0.7) |

Table 5: Results on 4 GLUE tasks using the popular LLM, LlaMA-13b. The fine-tuning is done in the few-data scenario (100 samples).

following three groups of experiments with open-sourced language models.

**Classification tasks**    Continuing the experiments in Table 1 and 4, we first experiment with the LlaMA-13b (Touvron et al., 2023) (13 billion parameters) on the SST-2, MNLI, RTE and Subj tasks. The results are presented in Table 5. We can see that by selecting proper prompt layers, our SPT method successfully help the LlaMA-13b backbone to achieve clear performance gains over the LPT baseline. In addition, we can see that LLM presents strong performances under the few-data settings. However, the LLMs requires much higher computational complexity and memory costs.

**Other English tasks**    We now also conduct experiments on other English tasks of different types: (a) COPA, a task focused on commonsense reasoning. (b) ShaRE-13, a nested named entity recognition (NER) task within the biomedical domain. (c) MultiArith, a task centered around arithmetic reasoning. To be consistent with Table 1, 4 and 5, we sample 100 samples from the training sets of these tasks as our training set. The hyper-parameter settings are the same with Table 1. The results of these experiments are presented in Table 6. From the above table, the following observations can be made: (1) ChatGPT demonstrates strong performance in many NLP tasks without fine-tuning. In contrast, when employing our SPT method with 100 training samples, fine-tuned LlaMA-2 13B exhibits either comparable or superior performance compared to ChatGPT. (2) On all the above tasks, when having comparable tunable parameters, SPT can outperform LoRA or LPT under the few-data setting.

### 5.8   Validating our SPT-DARTS method

In order to validate the effectiveness of our SPT-DARTS method, we now conduct two experiments. **Ablation on the hyper-network optimization method**    The first experiment is to substitute SPT-DARTS to DARTS (Liu et al., 2019a) or its variants

| Method | COPA (acc) | ShaRE-13 (f1) | MultiArith (acc) |
|---|---|---|---|
| ChatGPT | 0.732 | 0.331 | 0.953 |
| LlaMA-2 13B + LoRA | 0.718 | 0.532 | 0.888 |
| LlaMA-13b + LPT | 0.823 | 0.536 | 0.879 |
| LlaMA-13b + SPT | 0.836 | 0.553 | 0.907 |

Table 6: Results of fine-tuned LlaMA-13b on 3 tasks. The fine-tuning is done in the few-data scenario (100 samples).

P-DARTS (Chen et al., 2021), FairNAS (Chu et al., 2021) or $L_0$ regularization method (Louizos et al., 2017). The results is presented in Table 15 (Appendix H.1). We can see that our SPT method can obtain better learned SPT models than the other methods.

**SPT-DARTS on the NAS benchmark**    Note that the architectural consistency learning regularization of our SPT-DARTS method is generally applicable to neural architecture search. We now evaluate SPT-DARTS on the widely used NAS benchmark, NAS-benchmark-201 (Dong and Yang, 2020). We follow the same search setting as DARTS on NAS-benchmark-201. The results are in Table 16 of Appendix H.2. The results show that our SPT-DARTS can outperform ENAS and DARTS by a clear margin on NAS-benchmark-201.

## 6   Conclusion

In this work, we propose the SPT framework to automatically determine the optimal settings for prompt layers under the given PTM backbone and downstream task. We initialize a prompt hyper-network in which each layer has a prompt generator controlled by a learnable probabilistic gate. To better optimize the prompt hyper-network, we propose a novel SPT-DARTS method containing two novel modifications to the original DARTS' bi-level optimization process. Experiment results in full-data and few-shot scenarios demonstrate that SPT can achieve comparable or better performance than state-of-the-art PETuning methods while maintaining parameter and inference efficiency.

## Limitations

We showed that our proposed method can greatly improve the performance of prompt tuning on diverse NLU tasks and three different pre-trained models (i.e., RoBERTa-large, DeBERTa-large, and GPT2-large). However, the more large-scale pre-trained models with tens of billions or more parameters were not studied due to limited computation resources. In addition, a more comprehensive range of tasks, like text generation, should be investigated. Our framework can be easily transferred to other backbone architectures and different types of tasks. We are eager to validate our framework to a broader range of scenarios in future work.

## Ethics Statement

Our proposed method aims to improve prompt tuning in terms of performance under a budget of parameter efficiency. The datasets we experiment with are widely used in previous work and, to our knowledge, do not have any attached privacy or ethical issues.

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

## A    Pilot experiments for manually designed prompt inserting strategies

Prompt tuning (Lester et al., 2021) only adds a unified prompt at the word embeddings and does not perform well on downstream tasks. P-tuning v2 (Liu et al., 2021) improves upon prompt tuning by inserting prompts at each intermediate layer of PTM but suffers from difficulties in convergence. Liu et al. (2022a) argue that prompt tuning performs poorly mainly due to the long propagation path of task-related information, which causes the loss of task-related information during propagation in the frozen model and thus affects test performances. They further propose a late prompt tuning framework that brings significant improvements by generating an independent prompt for each sample and inserting these prompts at the 13-th layer of the RoBERTa-large model (Liu et al., 2019b). Since RoBERTa-large has 24 Transformer blocks, the prompts that carry task-related information still have to go through many transformer layers. Will another prompt generation layer after the 13-th layer help to preserve the task-related information and improve the test performances? Will adding more prompt generation layers be beneficial? Will adding too many prompt layers hurt the model performance?

Now we conduct pilot experiments on the RTE (Dagan et al., 2005) and Subj (Pang and Lee, 2004) datasets using the RoBERTa-large model to shed light on the above research questions. We consider the following model settings with RoBERTa-large as the backbone model:

- Model $\mathcal{M}_0$: following LPT (Liu et al., 2022a), we only insert soft prompt at the 13-th layer of PTM. The bottleneck dimension $r$ of the prompt generator is 128.

| Model | Tunable parameters | RTE (acc) | Subj (acc) |
|---|---|---|---|
| $\mathcal{M}_0$ | 263k | 76.4 (1.3) | 93.9 (0.2) |
| $\mathcal{M}_1$ | 262k | 76.7 (0.9) | 93.8 (0.2) |
| $\mathcal{M}_{2,1}$ | 295k | 75.1 (1.5) | 92.5 (0.2) |
| $\mathcal{M}_{2,2}$ | 295k | 75.6 (0.9) | 92.9 (0.4) |
| $\mathcal{M}_{2,3}$ | 295k | 76.4 (1.1) | 93.6 (0.1) |
| $\mathcal{M}_{2,4}$ | 295k | 76.8 (0.7) | 94.2 (0.2) |

Table 7: The experimental results for the pilot experiments, with the RoBERTa-large backbone. The evaluation metric for the RTE and Subj tasks is accuracy (acc).

- Model $\mathcal{M}_1$: different from LPT (Liu et al., 2022a), we set the 13-th and 19-th layers as the prompt layers. The bottleneck dimension $r$ of the prompt generators is set to 64 to maintain a comparable number of tunable parameters with model $\mathcal{M}_0$.

- Model $\mathcal{M}_{2,k}$: starting from layer 1, we set a prompt layer for every $k$ layers ($k = 1, 2, 3, 4$). The bottleneck dimension $r$ of the prompt generators are set to 6, 12, 18, and 24, respectively.

Note that in the above models, if a prompt layer has a prompt propagated from the previous layers, we will average the newly generated prompt with the old prompt and insert the resulting prompt into the prompt layer. We can see that the above models are simple variants of LPT (Liu et al., 2022a). The experimental settings for this pilot experiment are consistent with Section 5.2. The experiment results are reported in Table 7.

From Table 7, the following observations can be made: (a) we can see that some of the above simple manually designed models can perform comparably with or slightly outperform Liu et al. (2022a), with comparable numbers of tunable parameters. (b) Although adding more prompt layers can improve the performances (as for $\mathcal{M}_{2,4}$), setting too many prompt layers (like model $\mathcal{M}_{2,1}$) will hurt the model performances. The above observations suggest that we could obtain better performances of prompt tuning by appropriately setting prompt layers at certain intermediate layers.

## B    Datasets and evaluation metrics

**Dataset splits**    For SST-2, MNLI, MRPC, QNLI, QQP[3] and RTE datasets from the GLUE bench-

---

[3]https://www.quora.com/q/quoradata/

| Category | Datasets | \|train\| | \|dev\| | \|test\| | $\|\mathcal{Y}\|$ | Type | Labels |
|---|---|---|---|---|---|---|---|
| Single-sentence | SST-2 | 66349 | 1000 | 872 | 2 | sentiment | positive, negative |
| | MPQA | 7606 | 1000 | 2000 | 2 | opinion polarity | positive, negative |
| | MR | 7662 | 1000 | 2000 | 2 | sentiment | positive, negative |
| | Subj | 7000 | 1000 | 2000 | 2 | subjectivity | subjective, objective |
| | Trec | 4952 | 500 | 500 | 6 | question classification | abbr., entity, description, human, loc., num. |
| Sentence-pair | MNLI | 391702 | 1000 | 19647 | 3 | NLI | entailment, neutral, contradiction |
| | MRPC | 3668 | 204 | 204 | 2 | paraphrase | equivalent, not equivalent |
| | QNLI | 103743 | 1000 | 5463 | 2 | NLI | entailment, not entailment |
| | QQP | 362846 | 1000 | 40430 | 2 | paraphrase | equivalent, not equivalent |
| | RTE | 2490 | 138 | 139 | 2 | NLI | entailment, not entailment |

Table 8: The statistics of datasets evaluated in this work. For MNLI task, the number of samples in development and test sets is summed by matched and mismatched samples. $\|\mathcal{Y}\|$ is the number of classes for a dataset.

mark (Wang et al., 2018), the original test sets are not publicly available. Thus we follow Zhang et al. (2020) and Mahabadi et al. (2021) to construct train/dev/test splits as follows: (a) for datasets with fewer than 10k samples (RTE, MRPC), we divide the original validation set in half, using one half for validation and the other for testing. (b) for larger datasets, we split 1k samples from the training set as the development set and treat the original development set as the test set.

For four other classification datasets, we select a certain number of samples from the training set as the development set. The number of samples for each label is determined according to its proportion in the original training set. The dataset statistics after splitting are shown in Table 8.

**Metrics for the tasks** For MNLI, we report the average accuracy score on the matched and mismatched test sets. For MRPC and QQP, we report acc-f1, the average of the accuracy and F1 scores. And we report accuracy for all other tasks.

## C Implementation Details

**Pretrained backbones** We evaluate our method in both full-data and few-shot scenarios on three PTM backbones, RoBERTa-large (Liu et al., 2019b), DeBERTa-large (He et al., 2020), and GPT2-large (Radford et al., 2019).

**SPT Model settings** We use the HugginFace Transformers (Wolf et al., 2020) as the code base for implementing our method. The prompt length is 20. We follow Gao et al. (2021) and show the used manual templates and label words in Table 9 and Table 10, respectively. Note that since the vocabulary of the GPT2 model does not have the [MASK] token, we justly use it to represent the positions that are needed to predict. Unless otherwise specified, we set $\tau = 0.5$. That is, our method will keep the previous layer's prompt when inserting

new prompts. We also consider $\tau = 1$ for comparison. We set the bottleneck dimension $m = 128$ and the targeted number of prompt layers $K = 4$. For the PHM layers, we use the PyTorch implementation of Le et al. (2021), and set $n = 8$ for the main experiments (Tables 11 and 1). We also run models with $K \in \{1, 2, 8, 16\}$ for comparison.

**Settings for SPT-DARTS** For optimizing the hyper-network and learning the optimal prompt layer settings, we follow DARTS (Liu et al., 2019a)' bi-level optimization method. We equally split the original training set $\mathcal{D}_{train}$ into two splits, $\mathcal{D}_{\omega}$ and $\mathcal{D}_{\alpha}$ on epoch start. $\mathcal{D}_{\omega}$ is used to optimize the parameters in the prompt generators, and $\mathcal{D}_{\alpha}$ is used to optimize the gating parameters. The mean value $s$ for the Bernoulli architectural mask is set to 0.6. The learned SPT model is retrained from scratch in the original $\mathcal{D}_{train}$ set and evaluated on the dev and test sets. All the parameters are optimized with the AdamW (Loshchilov and Hutter, 2019) optimizer with linear learning rate decay, 6% warmup ratio, and learning rate 5e-4. Under the full-data scenario, the bi-level optimization and retraining procedures will run for $T$ epochs with batch size $B$. We set $T = 30$ and $B = 8$ for the MPQA, subj, TREC, MRPC, and RTE tasks, and $T = 15$ and $B = 64$ for SST-2 and QQP tasks, $T = 5$ and $B = 128$ for MNLI and QQP tasks. For the GPT2-large model, we use the gradient accumulation technique to avoid out-of-memory and control the actual batch size. Under the few-data scenario, the bi-level optimization and retraining procedures will run for 1000 steps with batch size $B = 8$. For the bi-level optimization, one step for the model parameters and one for the architectural parameters constitute a complete step. The bi-level optimization process will only run once. We report the average performances and standard deviations on the test set of the learned SPT model across 5

| Task | Template | Label words |
|------|----------|-------------|
| SST-2 | $\langle S_1 \rangle$ It was [MASK] . | positive: great, negative: terrible |
| MPQA | $\langle S_1 \rangle$ It was [MASK] . | positive: great, negative: terrible |
| MR | $\langle S_1 \rangle$ It was [MASK] . | positive: great, negative: terrible |
| Subj | $\langle S_1 \rangle$ It was [MASK] . | subjective: subjective, objective: objective |
| TREC | [MASK] : $\langle S_1 \rangle$ | abbreviation: Expression, entity: Entity, description: Description human: Human, location: Location, numeric: Number |
| MNLI | $\langle S_1 \rangle$ ? [MASK] , $\langle S_2 \rangle$ | entailment: Yes, netural: Maybe, contradiction: No |
| MRPC | $\langle S_1 \rangle$ [MASK] , $\langle S_2 \rangle$ | equivalent: Yes, not equivalent: No |
| QNLI | $\langle S_1 \rangle$ ? [MASK] , $\langle S_2 \rangle$ | entailment: Yes, not entailment: No |
| QQP | $\langle S_1 \rangle$ [MASK] , $\langle S_2 \rangle$ | equivalent: Yes, not equivalent: No |
| RTE | $\langle S_1 \rangle$ ? [MASK] , $\langle S_2 \rangle$ | entailment: Yes, not entailment: No |

Table 9: The manual templates and label words used on RoBERTa and DeBERTa models.

| Task | Template | Label words |
|------|----------|-------------|
| Subj | $\langle S_1 \rangle$ It was [MASK] . | subjective: subjective, objective: objective |
| TREC | [MASK] : $\langle S_1 \rangle$ | abbreviation: Expression, entity: Entity, description: Description human: Human, location: Location, numeric: Number |
| MRPC | $\langle S_1 \rangle$ $\langle S_2 \rangle$ They are [MASK] . | equivalent: Yes, not equivalent: No |
| RTE | $\langle S_1 \rangle$ $\langle S_2 \rangle$ They are [MASK] . | entailment: Yes, not entailment: No |

Table 10: The manual templates and label words used on the GPT2-large model.

random seeds under the full-data scenario and 10 random seeds under the few-shot scenario.

**Settings for the baselines** We also add manual templates in Tables 9 and 10 to transfer the downstream tasks to (masked) language modeling tasks. For adapter-based tuning methods, we set the down-projection size $m$ to 16. We set the soft prompt length to 20 for prompt tuning (Lester et al., 2021), P-tuning v2 (Liu et al., 2022b), IDPG (Wu et al., 2022) and LPT (Liu et al., 2022a). Besides, we set the down-projection size $m$ of IDPG (Wu et al., 2022) and LPT (Liu et al., 2022a) to 128. The hyperparameter $r$ and $\alpha$ in LoRA (Hu et al., 2021) are set to 8 and 16 on RoBERTa-large, 4 and 32 on GPT2-large. The settings for the optimizer, learning rate, warm-up, and batch size are the same as our SPT method. And we also report the average performances and standard deviations on the test set across 5 random seeds under the full-data scenario and 10 random seeds under the few-shot scenario.

## D Experimental results under the full-data setting

In the main content, we present the results for the few-shot scenario of 100 samples. Here, we present the results for the full-data scenario. The results are presented in Table 11.

## E Experimental results under few-shot settings

In the main content, we present the results for the few-shot scenario of 100 samples. Here, we present the results for the few-shot scenario of 200 samples and 500 samples. Under a given random seed, we randomly sample the training samples from the original training set. We will run the experiments over 10 different random seeds and report the mean and deviation on each task. The pretrained backbone model is RoBERTa-large. The results are presented in Table 12.

## F Efficiency evaluations for SPT

In this section, we measure the memory consumption and inference speed for three methods: prompt tuning, LPT and SPT. The pre-trained backbone is RoBERTa-large, the batch size is set to 32 and the maximum sequence length is 128. We report the measures in Table 13.

## G Effects of the prompt length $l$

We now present the experimental results for changing the prompt length in Table 14.

| Method | Tunable Params | SST-2 (acc) | MPQA (acc) | MR (acc) | Subj (acc) | TREC (acc) | MNLI (acc) | MRPC (acc-f1) | QNLI (acc) | QQP (acc-f1) | RTE (acc) | Avg |
|---|---|---|---|---|---|---|---|---|---|---|---|---|
| Model tuning | 355M | 95.4 (0.1) | 90.6 (0.5) | 90.5 (0.3) | 95.8 (0.4) | 93.6 (0.3) | 89.3 (0.1) | 89.5 (0.9) | 91.6 (0.3) | 90.7 (0.1) | 81.2 (1.0) | 90.8 |
| Adapter | 1.6M | 95.1 (0.2) | 89.2 (0.5) | 89.3 (0.4) | 95.0 (0.4) | 92.7 (0.3) | 88.5 (0.1) | 88.3 (1.0) | 91.0 (0.3) | 89.4 (0.7) | 78.8 (1.2) | 89.7 |
| AdapterDrop | 811K | 94.7 (0.3) | 89.1 (0.7) | 89.2 (0.5) | 94.0 (0.4) | 92.2 (0.5) | 88.3 (0.2) | 88.1 (1.3) | 90.4 (0.3) | 87.6 (0.3) | 77.6 (1.4) | 89.2 |
| BitFit | 273K | 94.9 (0.1) | 89.2 (0.9) | 89.9 (0.5) | 94.5 (0.1) | 92.6 (0.3) | 88.9 (0.1) | 87.7 (0.9) | 90.9 (0.2) | 87.9 (0.4) | 75.3 (1.1) | 89.2 |
| LoRA | 778K | 94.2 (0.3) | 89.9 (0.3) | 90.1 (0.5) | 94.8 (0.4) | 92.3 (0.6) | 88.8 (0.3) | 88.7 (0.6) | 91.1 (0.2) | 89.3 (0.3) | 78.5 (2.1) | 89.7 |
| Prompt Tuning | 21K | 93.9 (0.5) | 88.8 (0.8) | 88.6 (0.5) | 92.6 (0.6) | 90.4 (0.6) | 86.3 (0.4) | 78.3 (0.7) | 89.1 (0.1) | 81.2 (0.8) | 61.9 (0.5) | 85.1 |
| P-tuning v2 | 985K | 94.2 (0.4) | 89.9 (0.6) | 89.4 (0.4) | 93.2 (0.2) | 92.4 (0.6) | 88.5 (0.3) | 86.4 (2.1) | 89.5 (0.3) | 87.4 (0.2) | 69.1 (2.3) | 88.0 |
| IDPG | 296K | 94.3 (0.3) | 89.5 (0.6) | 89.6 (0.5) | 93.3 (0.6) | 91.9 (0.4) | 87.9 (0.5) | 87.9 (1.1) | 88.6 (0.4) | 86.3 (1.0) | 71.8 (1.9) | 88.2 |
| LPT | 263K | 94.8 (0.2) | 89.1 (0.3) | 89.6 (0.1) | 93.9 (0.2) | 92.1 (0.2) | 88.6 (0.3) | 88.3 (1.0) | 89.7 (0.5) | 88.5 (0.4) | 76.4 (1.3) | 89.1 |
| **Our SPT method** | | | | | | | | | | | | |
| SPT | 152K | 95.2 (0.1) | 90.5 (0.2) | 90.2 (0.2) | 95.3 (0.1) | 93.5 (0.2) | 89.0 (0.3) | 89.1 (0.7) | 90.8 (0.3) | 89.2 (0.4) | 79.2 (1.1) | 90.2 |
| SPT ($\tau = 1.0$) | 152K | 94.9 (0.2) | 89.9 (0.3) | 89.8 (0.4) | 94.7 (0.2) | 92.7 (0.2) | 88.6 (0.4) | 88.8 (1.0) | 90.2 (0.5) | 88.6 (0.4) | 78.3 (1.3) | 89.7 |

Table 11: The Overall comparison in full-data scenario. We report mean and standard deviation of performance over 5 different random seeds for all the methods. Bold and Underline indicate the best and the second best results. PT-256 indicates prompt tuning with prompt length 256. All the results are obtained using RoBERTa-large as the pre-trained backbone.

| Method | Tunable Params | SST-2 (acc) | MNLI (acc) | RTE (acc) | Subj (acc) |
|---|---|---|---|---|---|
| **Few-shot scenario: 200 samples** | | | | | |
| Model tuning | 355M | 88.9 (0.9) | 53.3 (2.1) | 51.6 (2.4) | 89.9 (1.1) |
| Prompt tuning | 26K | 88.2 (0.7) | 49.6 (1.7) | 58.1 (2.2) | 86.7 (1.2) |
| LPT | 329K | 90.9 (0.9) | 53.6 (1.5) | 59.4 (1.7) | 90.7 (0.9) |
| SPT | 186K | **92.3** (0.7) | **56.1** (1.6) | **61.6** (1.9) | **91.6** (1.0) |
| **Few-shot scenario: 500 samples** | | | | | |
| Model tuning | 355M | 91.0 (0.7) | 63.1 (1.8) | 57.2 (1.6) | 90.9 (0.7) |
| Prompt tuning | 26K | 89.1 (0.7) | 54.3 (2.1) | 63.5 (1.5) | 88.6 (1.1) |
| LPT | 263K | 91.1 (0.9) | 62.7 (1.6) | 66.8 (1.6) | 91.3 (0.8) |
| SPT | 152K | **92.6** (0.7) | **65.3** (1.4) | **68.1** (1.4) | **92.1** (0.9) |

Table 12: Results for the few-shot scenario of 200 samples and 500 samples. The pretrained backbone model is RoBERTa-large. Bold indicates the best result among the PETuning methods.

| Method | Speed (it/s) | Memory (GB) |
|---|---|---|
| Prompt tuning | 9.14 | 3.92 |
| LPT | 9.75 | 3.95 |
| SPT | 9.46 | 4.05 |

Table 13: Results for the efficiency measures. The memory and speed during inference on the RTE test set is presented. The backbone is RoBERTa-large, the batch size is set to 32 and maximum sequence length is 128.

# H  Appendix for analysis on the SPT-DARTS method

## H.1  Ablation on the NAS methods

Table 15 report the performance of SPT when the SPT-DARTS method is replaced by the other popular NAS methods: ENAS (Pham et al., 2018), DARTS (Liu et al., 2019a), P-DARTS (Chen et al., 2021), FairNAS (Chu et al., 2021), and $L_0$ regularization (Louizos et al., 2017). The model backbone is RoBERTa-large.

## H.2  Results on the NAS-Bench-201

NAS-Bench-201 (Dong and Yang, 2020) is the most widely applied and investigated NAS benchmark analyzing various NAS methods. NAS-Bench-201 provides a DARTS-like search space, containing 4 internal nodes with 5 associated operations. The search space consists of 15,625 architectures, with the ground truth performance of CIFAR-10, CIFAR-100 and ImageNet16-120 of each architecture provided. On NAS-Bench-201, the searching settings are kept the same as DARTS on (Dong and Yang, 2020). Table 16 reports the performances.

| Method | Prompt length $l$ | SST-2 (acc) | MNLI (acc) | RTE (acc) | Subj (acc) |
|---|---|---|---|---|---|
| LPT | 10 | 89.7 (0.8) | 52.5 (2.0) | 57.1 (3.5) | 89.7 (1.7) |
| LPT | 5 | 89.3 (0.9) | 52.0 (1.8) | 56.3 (2.9) | 88.9 (1.6) |
| LPT | 20 | 89.9 (0.8) | 52.8 (2.1) | 57.6 (3.3) | 89.9 (1.9) |
| SPT | 10 | 90.8 (1.0) | 54.9 (1.7) | 58.9 (2.3) | 90.8 (0.9) |
| SPT | 5 | 90.7 (0.9) | 54.7 (1.6) | 58.8 (2.1) | 90.8 (0.8) |
| SPT | 20 | 90.8 (1.1) | 54.8 (1.8) | 59.0 (2.3) | 90.9 (1.1) |

Table 14: Results on 4 GLUE tasks with different prompt length using RoBERTa-large as the backbone under the few-data scenario (100 samples). Bold indicates the best result among PETuning method.

| Method | Tunable Params | SST-2 (acc) | MNLI (acc) | RTE (acc) | Subj (acc) |
|---|---|---|---|---|---|
| SPT-DARTS | 152K | **95.2** (0.1) | **89.0** (0.3) | **79.2** (1.1) | **95.3** (0.1) |
| $L_0$ regularization | 152K | 94.1 (0.3) | 87.9 (0.7) | 78.1 (1.9) | 93.7 (0.4) |
| ENAS | 152K | 94.3 (0.2) | 88.2 (0.5) | 77.6 (1.7) | 94.3 (0.3) |
| DARTS | 152K | 94.2 (0.1) | 88.1 (0.5) | 78.3 (1.8) | 93.8 (0.4) |
| P-DARTS | 152K | 94.6 (0.2) | 88.4 (0.7) | 78.3 (1.6) | 94.5 (0.3) |
| FairNAS | 152K | 94.4 (0.3) | 88.2 (0.6) | 78.1 (1.8) | 94.3 (0.2) |

Table 15: Comparisons of different hyper-network optimization methods on our SPT framework.

| | costs (hours) | CIFAR-10 | | CIFAR-100 | | ImageNet16-120 | |
|---|---|---|---|---|---|---|---|
| | | valid | test | valid | test | valid | test |
| ENAS | 3.9 | 37.51 ± 3.19 | 53.89 ± 0.58 | 13.37 ± 2.35 | 13.96 ± 2.33 | 15.06 ± 1.95 | 14.84 ± 2.10 |
| DARTS | 3.2 | 39.77 ± 0.00 | 54.30 ± 0.00 | 15.03 ± 0.00 | 15.61 ± 0.00 | 16.43 ± 0.00 | 16.32 ± 0.00 |
| PC-DARTS | - | 89.96 ± 0.15 | 93.41 ± 0.30 | 67.12 ± 0.39 | 67.48 ± 0.89 | 40.83 ± 0.08 | 41.31 ± 0.22 |
| SPT-DARTS | 5.2 | 91.16 ± 0.45 | 93.89 ± 0.41 | 70.45 ± 0.52 | 70.61 ± 0.50 | 44.92 ± 0.48 | 45.23 ± 0.58 |

Table 16: Performance comparison on NAS-Bench-201 benchmark.