# OpenReview forum: "SPT: Learning to Selectively Insert Prompts for Better Prompt Tuning"
_EMNLP/2023/Conference — EMNLP 2023 Main_

### Official Review · Reviewer_yfyq · 2023-08-04

**Soundness:** 3

**Excitement:**

3: Ambivalent: It has merits (e.g., it reports state-of-the-art results, the idea is nice), but there are key weaknesses (e.g., it describes incremental work), and it can significantly benefit from another round of revision. However, I won't object to accepting it if my co-reviewers champion it.

**Paper Topic And Main Contributions:**

Existing parameter-efficient tuning framework, especially soft-prompt tuning, has not select the optimal prompt layer settings for downstream adaptations.This work discusses how to adaptively select the ideal prompt layers for task adaptions. Specifically, they inject a probabilistic gate per layer, and propose a bi-level optimization framework for their approach. Under few-shot and full-data settings, their framework does show competitive performance on prompting-based classification tasks.

**Questions For The Authors:**

See my questions in "reasons to accept", and "reasons to reject".

**Reasons To Accept:**

1. Their asked question is meaningful to the community, as "how do we find the optimal strategy of prompt injection, given the task at hand?". In PETuning (parameter-efficient tuning), it would be interesting to ask and answer this question from a theoretical view, even though this work is more from an empirical view.
2. For their designs, their optimization objective looks novel and interesting, combing and adapting existing work's key designs to their approach.
3. The experiments and discussions look convincing, and especially, their conclusion of "Transferability of the learned PG settings" is very interesting to me. I am really looking forward to hearing from you about explaining this phenomenon!

**Reasons To Reject:**

- Honestly, I do think such small modification is very incremental to the community (though still good questions), as you just focus on the simplest classification tasks. Right now, with very large language models, simple ICL (in-context learning) performance is already very strong. Therefore, it would be very good to show your advantages on other use cases w/ LMs or even other models.

**Reproducibility:**

4: Could mostly reproduce the results, but there may be some variation because of sample variance or minor variations in their interpretation of the protocol or method.

**Reviewer Confidence:**

4: Quite sure. I tried to check the important points carefully. It's unlikely, though conceivable, that I missed something that should affect my ratings.

---

> ### Author Rebuttal · Authors · 2023-08-28
>
> We express our gratitude to the reviewer for conducting a comprehensive review. The insightful questions and suggestions provided will be carefully considered as we work on the final version.
>
>
> ### Question 1
>
> QUESTION: Whether our method works for other use cases w/ LMs or even other models?
>
>
> We thank the reviewers for the great question. We will use experiments to show that our method works well with the recent released LLMs, on a wide range of tasks. Notably, after fine-tuning, the open-sourced models will perform better than ChatAGPT or GPT-4 with ICL.
>
>
> #### (1a) Results of English tasks
>
> First, we experimented with English tasks. In the case of ChatGPT, we provided as many demonstrations as possible to fill its 4096-context length. During the fine-tuning process, we utilized the recently released open-source LLM, LlaMA-2 13B by Meta (available at: https://huggingface.co/meta-llama/Llama-2-13b-hf) as the backbone model. This model was fine-tuned using a few-data approach, utilizing only 200 samples.
>
> For our SPT method, we generated soft prompts by utilizing the hidden states of the instructions (or prompts). These soft prompts were then concatenated after the instruction. We retained a total of 8 prompt layers, each with a length of 10 tokens. The bottleneck dimension was set at 512, and the rank 'n' was set to 128. With this configuration, the number of tunable parameters amounted to 11.5 million.
>
> In the case of LoRA method, we set the rank to 4, and this rank was applied to all linear layers within the transformer block. This resulted in a total of 15.6 million tunable parameters.
>
> The experiments were conducted on the following tasks: (a) RTE, which involves natural language inference. (b) COPA, a task focused on commonsense reasoning. (c) ShaRE-13, a nested named entity recognition (NER) task within the biomedical domain. (d) MultiArith, a task centered around arithmetic reasoning. The results of these experiments are presented in the following table.
>
>
> | Task      | ChatGPT      |  LlaMA-2 13B + LoRA      |   LlaMA-2 13B + SPT      |
> |-------------|--------|--------|--------|
> | RTE        |   0.848   |  0.857   |   0.869   |
> | COPA        |   0.732   |  0.887   |   0.906   |
> | ShaRE-13        |   0.331   |  0.743   |   0.764   |
> | MultiArith     |   0.953   |  0.956   |   0.959   |
>
> From the above table, the following observations can be made:
> - (1) ChatGPT demonstrates strong performance in tasks such as RTE and MultiArith. In contrast, when employing our SPT method with 200 training samples, fine-tuned LlaMA-2 13B exhibits either comparable or superior performance compared to ChatGPT.
> - (2) Across the ShaRE-13 and COPA tasks, LlaMA-2 13B showcases a clear performance advantage over ChatGPT.
> - (3) On all the above tasks, when having comparable tunable parameters, SPT can outperform LoRA under the few-data setting.
>
> The outcomes of these experiments underscore the effectiveness of SPT in enhancing the performance of an open-source LLM, enabling it to either match or surpass ChatGPT's performance across diverse tasks.
>
>
>
> #### (1b) Results of Chinese tasks
>
> For Chinese tasks, we choose the CMeEE-v2 (medical NER), CMeIE (medical triple extraction task), and IMCS-V2-MRG (medical report generation) tasks from PromptCBLUE (https://tianchi.aliyun.com/competition/entrance/532085/introduction). The backbone LLM is Chinese LlaMA-2 13B (https://huggingface.co/ziqingyang/chinese-llama-2-13b/tree/main) The other experimental settings are kept the same with above.
>
> | Task      | ChatGPT      |  Chinese LlaMA-2 13B + LoRA      |   Chinese LlaMA-2 13B + SPT      |
> |-------------|--------|--------|--------|
> | CMeEE-v2        |   0.469   |  0.673   |   0.687   |
> | CMeIE        |   0.305   |  0.602   |   0.618   |
> | IMCS-V2-MRG        |   0.325   |  0.474   |   0.486   |
>
> From the results, we observe the following: (a) Fine-tuned Chinese LlaMA-2 13B outperforms ChatGPT significantly in the context of the two Chinese medical information extraction tasks. (b) While ChatGPT demonstrates strong performance in open-domain summarization, its performance in IMCS-V2-MRG is notably inferior to that of fine-tuned Chinese LlaMA-2 13B in terms of Rouge-L score. This task necessitates the model to succinctly summarize patient-doctor dialogues, and Language Models (LLMs) exhibit superior summarization capabilities in this style following fine-tuning.
>
>
>
> ### Question 2
>
> QUESTION: On the contributions of the work.
>
> We believe that our work is a non-trivial addition to the current literature:
> - Our approach, called SPT, focuses on the automatic selection of the prompt layer in pre-trained transformer models. This is achieved by optimizing the learnable probabilistic gate $a_i$, in conjunction with the parameters in the prompt generator. To enhance the optimization of the learnable probabilistic gate $a_i$, we introduce the re-parameterization method and architectural consistency regularization techniques.
> - Through experiments on a series of classification tasks, commonsense QA, information extraction tasks and medical report summarization tasks, we demonstrate that our SPT method performs well with different pretrained models, including the recently released LlaMA-2 models. Our method consistently achieves superior fine-tuning performance compared to LoRA, especially when working with limited training data.

---

### Official Review · Reviewer_XNTB · 2023-08-05

**Typos Grammar Style And Presentation Improvements:** 1. I'm wondering the usage of bracket…
**Soundness:** 3

**Excitement:**

3: Ambivalent: It has merits (e.g., it reports state-of-the-art results, the idea is nice), but there are key weaknesses (e.g., it describes incremental work), and it can significantly benefit from another round of revision. However, I won't object to accepting it if my co-reviewers champion it.

**Paper Topic And Main Contributions:**

This study presents the method for selecting specific layers to input a prompt with a learnable probabilistic gate. Through the bi-level optimization, they were able to enhance the convergence of the learnable gates. Such improvement results in enhanced overall performance. Numerous experiments have been conducted to demonstrate the efficacy of this framework.

**Questions For The Authors:**

Please refer to the reasons to reject section.

**Reasons To Accept:**

1. The researchers propose a technique for identifying the optimal layer for prompt integration. Instead of utilizing an empirical selection process, this approach employs an algorithmic search for finding the most effective layer.

2 .To add clarity and depth to their findings, the authors provide a comprehensive analysis from various perspectives.

3. The proposed method outperforms the prior related research.

**Reasons To Reject:**

1. Based on my understanding, the proposed method appears to adaptively update the prefix using a learnable probabilistic gate. However, the provided figure seems to depict a process of concatenating the prompt instead. To ensure the illustration aligns with the method's description, I suggest revising the figure.

2. With regards to training, I'm curious about the requirements for the prompt generator. Is it necessary to have a separate prompt generator for each layer? Or, could we train just one prompt generator, with the differing input hidden states provided from layer to layer?

3. Why the prompt length for the proposed method and other comparing methods are set differently? (searched as hyperparameter?)

4. If equations 3 and 8 are accurately bracketed, it implies that you’re highly weighting the previous prompt at the outset of the training. It would be beneficial to include a discussion or analysis showing the progression of the learnable gates throughout the training. This would provide valuable insights into how the previous prompts are integrated over time.

5. I'm interested in how the model performs when consistency regularization is not applied. If reasonable performance can be achieved without the use of this regularization technique, it could potentially allow for a trade-off between performance and computational cost. Specifically, this might eliminate the need for multiple forward passes, thereby reducing computational demand.


**Reproducibility:**

3: Could reproduce the results with some difficulty. The settings of parameters are underspecified or subjectively determined; the training/evaluation data are not widely available.

**Reviewer Confidence:**

4: Quite sure. I tried to check the important points carefully. It's unlikely, though conceivable, that I missed something that should affect my ratings.

---

> ### Author Rebuttal · Authors · 2023-08-28
>
> We express our gratitude to the reviewer for conducting a comprehensive review. The insightful questions and suggestions provided will be carefully considered as we work on the final version.
>
>
> ### Question 1
>
> QUESTION: To ensure the illustration aligns with the method's description, I suggest revising the figure.
>
> Response: We will adjust Figure 2 to more accurately depict our method, particularly emphasizing Equation 3.
>
>
> ### Question 2
>
> QUESTION: Could we train just one prompt generator, with the differing input hidden states provided from layer to layer?
>
> Response: This is a thought-provoking question. The configuration of the shared prompt generator, referred to as SPT-shared-PG, is definitely worth investigating. We now conduct experiments to compare SPT-shared-PG with our main model, SPT.
>
> #### (2a) Experiments with RoBERTa-large
>
> We first conduct experiments on the RoBERTa-large backbone, same as Table 1. For SPT-shared-PG, the number of finally kept prompt layers are 4, and the prompt length is 10, and bottleneck dimension is 128, and rank n = 32. Under this setting, the tunable parameter number is 152K, the same with SPT on Table 1. In the few-shot scenario of 100 training samples, the avg score on the 10 classification task is 77.2, while our main model, SPT, achieve the avg score of 78.0. In the few-shot scenario of 200 training samples, the avg score on the 10 classification task is 77.8, while our main model, SPT, achieve the avg score of 78.5.
>
>
> We initially conducted experiments using the RoBERTa-large backbone. For the SPT-shared-PG model, we retained a total of 4 prompt layers, with each prompt having a length of 10 tokens. The bottleneck dimension was set to 128, and the rank 'n' was chosen as 32. With these parameters, the number of tunable parameters was 152K, which aligns with the SPT configuration in our Table 1.
>
> In the scenario where only 100 training samples were available, SPT-shared-PG attained an average score of 77.2 across the 10 classification tasks. In contrast, our primary model, SPT, achieved a slightly higher average score of 78.0. As the number of training samples increased to 200 in the few-shot scenario, the average score of SPT-shared-PG for the 10 classification tasks improved to 77.8. Once again, our primary model, SPT, exhibited superior performance with an average score of 78.5.
>
>
>
> #### (2b) Experiments with open-sourced LLMs
>
>
> We now conduct experiments using the recently released open-source language model, LlaMA-2 13B, developed by Meta (available at: https://huggingface.co/meta-llama/Llama-2-13b-hf). We perform fine-tuning with a limited dataset comprising 200 samples. Our experiments cover the following tasks: a) Recognizing Textual Entailment (RTE) - a natural language inference task. b) Choice of Plausible Alternatives (COPA) - a task involving commonsense reasoning. c) ShaRE-13 - a nested Named Entity Recognition (NER) task within the biomedical domain. d) MultiArith - a task focused on arithmetic reasoning.
>
>
>  For SPT, the soft prompts are generated using the hidden states of the instructions (or prompts), and they are concatenated after the instruction. Specifically, eight prompt layers are retained, each with a prompt length of ten. The bottleneck dimension is set at 512, and the rank, denoted as 'n', is 128. With these settings, the total number of tunable parameters is 11.5 million. For SPT-shared-PG, the experimental setup is similar, except for setting 'n' to 512 to ensure comparability in the number of tunable parameters.
>
>  The results are reported below. SPT-shared-PG performs slightly better than LoRA under the few-data setting of 200 samples, but it underperforms compared to our main model, SPT.
>
>
>
> | Task         |  LlaMA-2 13B + LoRA      |   LlaMA-2 13B + SPT      | LlaMA-2 13B + SPT-shared-PG      |
> |-------------|--------|--------|--------|
> | RTE        |    0.857   |   0.869   |   0.857   |
> | COPA        |    0.887   |   0.906   |   0.901   |
> | ShaRE-13        |    0.743   |   0.764   |   0.756   |
> | MultiArith     |    0.956   |   0.959   |   0.957   |
>
>
>
> From the results presented above, it is evident that the configuration of SPT-shared-PG demonstrates commendable performance across various tasks. However, our primary model, SPT, outperforms it. This outcome is to be expected, as we employ distinct prompt generators to align with the hidden states of different layers.
>
>
> ### Question 3
>
> QUESTION: Why the prompt length for the proposed method and other comparing methods are set differently?
>
> Response:  We apologize for any confusion caused by the typo. In line 457, the prompt length for all the baseline prompt-tuning-based methods is actually 10, not 20.
>
> As stated in line 559, we configured the prompt length in the main experiment (Table 1) to be 10 in order to ensure a fair comparison. Additionally, the outcomes of various prompt lengths are provided in Table 13 of Appendix I.
>
>
> ### Question 4
>
> QUESTION: It would be beneficial to include a discussion or analysis showing the progression of the learnable gates throughout the training.
>
> Response:  We thank the reviewer for this great question. In the final version, we intend to incorporate a visual representation illustrating the dynamic changes of the learnable gates throughout the course of training.
>
> Based on our observations, we typically notice fluctuations in the values of $a_i$ during the initial steps of training. After around 200 steps, we observe a smoother convergence of $a_i$ towards either 0 or 1.
>
>
> ### Question 5
>
> QUESTION: If equations 3 and 8 are accurately bracketed, it implies that you’re highly weighting the previous prompt at the outset of the training.
>
>
> Response:  Actually, in equation 3, we use the hyper-parameter $\tau \n {0.5,1.0}$ to determine whether the prompt layer will incorporate the soft prompt propagated from the previous layer. We now explain how will $\tau$ the final learned SPT model (where a_i is discretized to 0 or 1)
> - If $\tau=0.5$,
> 	- at the prompt layer, a_i=1, so the soft prompt will be the average of the previous layer’s soft prompt and the newly injected soft prompt.
> 	- At other layers, a_i=0, so that no new soft prompt will be included. And this layer will use the soft prompt propagated from the previous layer (or no soft prompt at all).
> - If $\tau=1.0$,
> 	- At the prompt layer, $a_i=1$, so the soft prompt will be only the newly injected soft prompt.
> 	- At other layers, $a_i=0$, so that no new soft prompt will be included. And this layer will use the soft prompt propagated from the previous layer (or no soft prompt at all).
>
>
> ### Question 6
>
> QUESTION: how the model performs when consistency regularization is not applied.
>
>
> Response:  In Section 5.7 (lines 621 to 631), we conducted an ablation study on the hyper-network optimization method introduced in Section 4.4. This method involves the re-parameterization of probabilistic gates and architectural consistency learning. The results, presented in Table 14 of Appendix J, demonstrate that our approach achieves better performance compared to other hyper-network optimization methods.
>
> As the reviewer pointed out, we haven’t conducted an ablation study solely on consistency regularization. In the following table, we present the results. 'SPT w/o consistency' refers to conducting the hyper-network optimization method without the consistency regularization objective. The other settings remain the same as the responses for Question 2.
>
>
> | Task         |  LlaMA-2 13B + LoRA      |   LlaMA-2 13B + SPT      | SPT w/o consistency      |
> |-------------|--------|--------|--------|
> | RTE        |    0.857   |   0.869   |   0.863   |
> | COPA        |    0.887   |   0.906   |   0.903  |
> | ShaRE-13        |    0.743   |   0.764   |   0.759   |
> | MultiArith     |    0.956   |   0.959   |   0.956   |
>
>
> We observe that when utilizing the LlaMA-2 13B backbone along with a limited dataset of 200 samples for fine-tuning, the inclusion of the consistency regularization term results in notable performance improvements. However, it's noteworthy that even in the absence of consistency regularization, SPT consistently outperforms the LoRA method. Therefore, if computational resources permit, it would be advisable to incorporate the consistency regularization technique for enhanced training outcomes.

---

### Official Review · Reviewer_AMiu · 2023-08-09

**Typos Grammar Style And Presentation Improvements:** There seems a missing citation in lin…
**Soundness:** 4

**Excitement:**

4: Strong: This paper deepens the understanding of some phenomenon or lowers the barriers to an existing research direction.

**Missing References:**

It seems that the citation to P-tuning (Liu et al., 2022b) in line 131 should be replaced to [1].

References:

[1] Liu, X., Zheng, Y., Du, Z., Ding, M., Qian, Y., Yang, Z., & Tang, J. (2021). GPT understands, too. arXiv preprint arXiv:2103.10385.

**Paper Topic And Main Contributions:**

This paper studies the selection of prompt layer in prompt tuning methods. The authors first conduct a pilot exam to show that prompt inserting strategies affects the performance of prompt tuning. The authors further propose an optimization method to automatically search for the optimal layer to insert prompt. Results of experiments shows the effectiveness of the proposed method.

**Questions For The Authors:**

A: At training stage, will all the $a_i$ be updated? If yes, it seem that SPT actually need $l$ prompt generators, where $l$ is the number of layers of the language model. If that is the case, then SPT still need to train $l$ prompt generators since SPT-DARTS first optimize $\omega$ on $\mathcal{D}_{\omega}$.

B: How is the hyper-parameter choosed for baseline models?

C: Do the author(s) think the method or idea proposed in this paper can directly facilitate or have any implications on the usage and adaption of cutting-edge large language models (e.g. ChatGPT)?

**Reasons To Accept:**

1. This paper proposes a novel prompt tuning approach, that is, automatically selecting several layers to insert instance-dependent soft prompts. This paper also proposes the corresponding optimization method.
2. The results of the experiments verify the effectiveness of proposed method.
3. The ablation study is comprehensive.

**Reasons To Reject:**

1. The writing in section 4.3 and 4.4 is confusing. If my understanding is right, I suggest the author add a subsection in section 4 illustrating that SPT uses only the layers with larger $a_i$ for prompt inserting.
2. Given the emergent of cutting-edge large language models and their decent in-context performance, the technique proposed in this paper seems to be redundant and outdated.

**Reproducibility:**

4: Could mostly reproduce the results, but there may be some variation because of sample variance or minor variations in their interpretation of the protocol or method.

**Reviewer Confidence:**

3: Pretty sure, but there's a chance I missed something. Although I have a good feel for this area in general, I did not carefully check the paper's details, e.g., the math, experimental design, or novelty.

---

> ### Author Rebuttal · Authors · 2023-08-28
>
> We appreciate the thorough reviews provided by the reviewer. We will carefully consider the valuable questions and suggestions as we prepare the final version.
>
>
> ### Question 1
>
> QUESTION: Given the emergent of cutting-edge large language models and their decent in-context performance, the technique proposed in this paper seems to be redundant and outdated.
>
> Response: We hold a different perspective. OpenAI's recent announcement on August 22, 2023, reveals the availability of Fine-tuning for GPT-3.5 Turbo, followed by fine-tuning support for GPT-4 later this fall. (URL: https://openai.com/blog/gpt-3-5-turbo-fine-tuning-and-api-updates) OpenAI explains that fine-tuning offers distinct advantages compared to prompting or in-context learning: (a) Enhanced steerability. (b) Improved adherence to specific formats. (c) Customizable tone. (d) Reduced prompt length, leading to lower API fees, enhanced efficiency, and decreased GPU memory usage.
>
> In the following, we will present experimental evidence demonstrating that even language models weaker than ChatGPT or GPT-4 can outperform these models when fine-tuned on a diverse set of tasks.
>
>
> #### (1a) Results of English tasks
>
> First, we experimented with English tasks. In the case of ChatGPT, we provided as many demonstrations as possible to fill its 4096-context length. During the fine-tuning process, we utilized the recently released open-source LLM, LlaMA-2 13B by Meta (available at: https://huggingface.co/meta-llama/Llama-2-13b-hf) as the backbone model. This model was fine-tuned using a few-data approach, utilizing only 200 samples.
>
> For our SPT method, we generated soft prompts by utilizing the hidden states of the instructions (or prompts). These soft prompts were then concatenated after the instruction. We retained a total of 8 prompt layers, each with a length of 10 tokens. The bottleneck dimension was set at 512, and the rank 'n' was set to 128. With this configuration, the number of tunable parameters amounted to 11.5 million.
>
> In the case of LoRA method, we set the rank to 4, and this rank was applied to all linear layers within the transformer block. This resulted in a total of 15.6 million tunable parameters.
>
> The experiments were conducted on the following tasks: (a) RTE, which involves natural language inference. (b) COPA, a task focused on commonsense reasoning. (c) ShaRE-13, a nested named entity recognition (NER) task within the biomedical domain. (d) MultiArith, a task centered around arithmetic reasoning. The results of these experiments are presented in the following table.
>
>
> | Task      | ChatGPT      |  LlaMA-2 13B + LoRA      |   LlaMA-2 13B + SPT      |
> |-------------|--------|--------|--------|
> | RTE        |   0.848   |  0.857   |   0.869   |
> | COPA        |   0.732   |  0.887   |   0.906   |
> | ShaRE-13        |   0.331   |  0.743   |   0.764   |
> | MultiArith     |   0.953   |  0.956   |   0.959   |
>
> From the above table, the following observations can be made:
> - (1) ChatGPT demonstrates strong performance in tasks such as RTE and MultiArith. In contrast, when employing our SPT method with 200 training samples, fine-tuned LlaMA-2 13B exhibits either comparable or superior performance compared to ChatGPT.
> - (2) Across the ShaRE-13 and COPA tasks, LlaMA-2 13B showcases a clear performance advantage over ChatGPT.
> - (3) On all the above tasks, when having comparable tunable parameters, SPT can outperform LoRA under the few-data setting.
>
> The outcomes of these experiments underscore the effectiveness of SPT in enhancing the performance of an open-source LLM, enabling it to either match or surpass ChatGPT's performance across diverse tasks.
>
>
>
> #### (1b) Results of Chinese tasks
>
> For Chinese tasks, we choose the CMeEE-v2 (medical NER), CMeIE (medical triple extraction task), and IMCS-V2-MRG (medical report generation) tasks from PromptCBLUE (https://tianchi.aliyun.com/competition/entrance/532085/introduction). The backbone LLM is Chinese LlaMA-2 13B (https://huggingface.co/ziqingyang/chinese-llama-2-13b/tree/main) The other experimental settings are kept the same with above.
>
> | Task      | ChatGPT      |  Chinese LlaMA-2 13B + LoRA      |   Chinese LlaMA-2 13B + SPT      |
> |-------------|--------|--------|--------|
> | CMeEE-v2        |   0.469   |  0.673   |   0.687   |
> | CMeIE        |   0.305   |  0.602   |   0.618   |
> | IMCS-V2-MRG        |   0.325   |  0.474   |   0.486   |
>
> From the results, we observe the following: (a) Fine-tuned Chinese LlaMA-2 13B outperforms ChatGPT significantly in the context of the two Chinese medical information extraction tasks. (b) While ChatGPT demonstrates strong performance in open-domain summarization, its performance in IMCS-V2-MRG is notably inferior to that of fine-tuned Chinese LlaMA-2 13B in terms of Rouge-L score. This task necessitates the model to succinctly summarize patient-doctor dialogues, and Language Models (LLMs) exhibit superior summarization capabilities in this style following fine-tuning.
>
>
> In the final version, we will also present the results of GPT-4 on these tasks. And we will also try out the OpenAI’s fine-tuning APIs to see how ChatGPT performs after fine-tuning with the same set of 200 training samples.
>
>
>
> ### Question 2
>
> QUESTION: Will $a_i$ be updated?
>
> Response:  Yes, the learnable gate parameters $a_i$ will be updated during training following our equation 4 and equation 7.
> - Through optimization, the probabilistic gate $a_i$’s value will move toward 0 or 1, acting as importance scores for the prompt layers. Then the prompt generators with the $K$ largest $a_i$ values will be kept, while other prompt generators will be discarded. And the gate parameters $a_i$ will be set to 1 for those layers with prompt generators, and 0 for other layers.
> - Then the learned SPT model with the selected K prompt generators will be trained on the training set again, following the standard model training procedure.
> - Thus, during inference, the learned SPT model will only has K prompt generators.
>
>
> ### Question 3
>
> QUESTION: I suggest the author add a subsection in section 4 illustrating that SPT uses only the layers with larger $a_i$ for prompt inserting.
>
> Response:  We will take the time to thoroughly revise Section 4 to ensure that it is more reader-friendly.
>
> If K prompt generators are selected, then in the transformer layer without a prompt generator, we will utilize the soft prompt from the preceding layer, or in some cases, no soft prompt at all. When the current transformer layer contains a prompt generator, it combines the soft prompt from the preceding layer (or no previous soft prompt) with the newly generated soft prompt. Our Figure 3 (on page 7) demonstrates the K selected prompt layers on the RoBERTa-large backbone.
>
>
> ### Question 4
>
> QUESTION: How is the hyper-parameter choosed for baseline models?
>
>
> Response:  For the baseline comparisons, we implement the baseline methods using OpenDelta (https://github.com/thunlp/OpenDelta) or their official code repositories. With these methods, we only adjust the hyperparameters directly associated with tunable parameter quantities (such as rank, bottleneck dimension, and prompt length). This ensures that the tunable parameter quantities align with those presented in Table 1.
>
> As for the other hyperparameters, particularly the training-related ones, we determine their values using the following approach: If OpenDelta is utilized, we adopt the parameter settings outlined in its corresponding paper, 'Delta tuning' (https://arxiv.org/abs/2203.06904). In cases where the original code of a method is employed, we adhere to the hyperparameters suggested by the original authors.
>
>
>
> ### Question 5
>
> QUESTION: Do the author(s) think the method or idea proposed in this paper can directly facilitate or have any implications on the usage and adaption of cutting-edge large language models (e.g. ChatGPT)?
>
>
> Response:  Despite the fact that ChatGPT's fine-tuning method is controlled by ChatGPT, we believe that our method can be applied to fine-tune the more recent LLMs. In the response above to question 1, we have presented some of our recent experimental results with the open-sourced LlaMA-2 models.
>
> In the final version, we will present more comprehensive experimental results using various open-source LLMs such as ChatGLM-6B, Baichuan-13B, and Falcon, among others.
>
>
>
>
> ### Question 6
>
> QUESTION: On the missing References.
>
> Response:  We will correct this error in the final version. Additionally, we will thoroughly review all references to ensure the proper citations have been made.
>
>
> ### Question 7
>
> QUESTION: On Typos Grammar Style And Presentation Improvements.
>
> Response:  We will thoroughly proofread the entire manuscript and enhance the presentation of this work in the final version.

---

### Meta-Review · Area_Chair_6hPq · 2023-09-16

**Recommendation:** 4

**Metareview:**

This paper proposes an approach for learning to select prompt tuning layers. The authors formulate this problem as a bi-level optimization problem, and proposes an approach for learning gating functions which modulate the layer selection procedure. Experiments show that the proposed approach is able to outperform various baselines across standard datasets. During the discussion period, the authors have performed extensive additional experiments to address many of the reviewers' concerns. This paper is a solid contribution to the parameter-efficient learning literature.

[As a side note, some reviewers raised concerns that this paper may be outdated or irrelevant given the the ICL + LLM paradigm with closed source models. I do not share this viewpoint, and have not given much weight to these concerns in the decision process.]

---

### Decision · Program_Chairs · 2023-10-07

**Decision:**

Accept-Main

**Comment:**

This paper proposes an approach for learning to select prompt tuning layers. The authors formulate this problem as a bi-level optimization problem, and proposes an approach for learning gating functions which modulate the layer selection procedure. Experiments show that the proposed approach is able to outperform various baselines across standard datasets. During the discussion period, the authors have performed extensive additional experiments to address many of the reviewers' concerns. This paper is a solid contribution to the parameter-efficient learning literature.

[As a side note, some reviewers raised concerns that this paper may be outdated or irrelevant given the the ICL + LLM paradigm with closed source models. I do not share this viewpoint, and have not given much weight to these concerns in the decision process.]